# In Vitro Infection Model Using A6 Cells Sets the Stage for Host–*Batrachochytrium salamandrivorans* Exploration

**DOI:** 10.3390/jof11020156

**Published:** 2025-02-18

**Authors:** Elin Verbrugghe, Frank Pasmans, An Martel

**Affiliations:** Wildlife Health Ghent, Department of Pathobiology, Pharmacology and Zoological Medicine, Faculty of Veterinary Medicine, Ghent University, B-9820 Merelbeke, Belgium; frank.pasmans@ugent.be (F.P.); an.martel@ugent.be (A.M.)

**Keywords:** *Batrachochytrium salamandrivorans*, amphibian chytrid fungus, host–pathogen interactions, in vitro infection tool, A6 cells

## Abstract

The chytrid fungus *Batrachochytrium salamandrivorans* (Bsal) poses a significant threat to amphibian biodiversity, driving severe declines in salamander populations in Europe. While understanding the host–pathogen interaction may yield novel avenues for disease mitigation, effective in vitro models are currently lacking. We here develop a cell-culture-based model using A6 cells to reproduce the complete life cycle of Bsal in vitro, encompassing key stages such as β-galactose-associated cell attachment, active host cell penetration, intracellular maturation, host cell death, and Bsal release. Using imaging techniques, we provide evidence that Bsal penetrates A6 cells through a mechanism independent of conventional host actin dynamics. Our comparative analysis reveals that Bsal infection closely mirrors responses observed in native salamander skin tissues, validating the A6 cell line as an effective surrogate for in vivo studies. This research enhances our understanding of Bsal’s pathogenicity and emphasizes the potential of the A6 cell model for future studies.

## 1. Introduction

Amphibians continue to face a persistent and escalating threat from emerging infectious diseases, exemplified by the devastating impact of chytridiomycosis caused by the chytrid fungi *Batrachochytrium dendrobatidis* (Bd) and *Batrachochytrium salamandrivorans* (Bsal) [1]. In 2013, Bsal was identified as a new chytrid species, able to infect vertebrates and particularly threatening salamander populations [2]. Bsal has since been linked to severe population declines and extinctions, underscoring the urgent necessity for the development and implementation of effective strategies to comprehend and mitigate its impact [2,3]. The current lack of sustainable long-term mitigation measures stresses the importance of continuing to expand our knowledge of the host–pathogen interaction.

Traditionally, elucidating the pathogenesis and host–pathogen interactions of Bd and Bsal has predominantly relied on in vivo trials utilizing live animals [4,5]. These studies offer a comprehensive understanding of disease progression within the complex physiological and immunological context of the host organism. In vivo models are particularly advantageous for studying species-specific and individual variations in responses to Bsal infection, as they account for factors such as amphibian immunity and environmental conditions that play a crucial role in disease development [6,7]. They provide a more holistic understanding, and they also allow for the evaluation of long-term effects.

However ethical considerations surrounding animal experimentation emphasize the imperative of exploring alternative methodologies to dissect these intricate interactions. In this context, in vitro and ex vivo systems emerge as promising avenues for dissecting host–pathogen dynamics while reducing reliance on live animal experimentation. Several alternatives to in vivo infection trials have been proposed. Some of these alternatives, such as those utilizing nematodes (*Caenorhabditis elegans*) [8], mucosomes [9,10], or bacteria [11,12,13], focus on specific components of the skin or host–pathogen interaction. While these models provide valuable insights into specific aspects of Bd and Bsal interactions, they do not replicate the intracellular nature of Bd and Bsal within the amphibian epidermis, which is crucial for understanding the full host–pathogen relationship. Among models that better replicate the epidermal environment, approaches involving full-thickness explants (FTE), stripped epidermal explants (SEE), or primary keratinocytes derived from the frogs *Xenopus* (*X*.) *laevis*, *X. tropicalis*, or *Litoria caerulea* have been developed in Bd research [14,15]. These systems closely recapitulate the attributes and functionalities of their tissue of origin, thereby representing aspects of in vivo biology. However, they still require the use of animals, and these models have a constrained lifespan.

An alternative approach has recently emerged that circumvents the need for animal experimentation by leveraging a continuous amphibian cell line, A6 cells [15,16,17]. Derived from the African clawed frog (*X. laevis*), A6 cells serve as a pragmatic model system that faithfully replicates key facets of the initial host–Bd interaction [15]. Continuous cell lines, due to their adaptability to in vitro conditions, offer enhanced practicality in experimental manipulation and prolonged maintenance in culture. Additionally, their scalability allows for rapid expansion of cell populations as needed. Recent discoveries underscore Bsal’s affinity for β-galactose ligands [18] and although A6 cells originate from a Bsal-resistant host, the presence of the Bsal receptor on these cells could render them a viable alternative.

In this study, we aim to evaluate the use of A6 cells as a tool to study Bsal–host interactions in vitro. We detail the processes of Bsal invasion, including actin rearrangements, and describe its life cycle within A6 cells. Furthermore, we compare these findings with the in vivo and ex vivo infection responses observed in salamander skin tissue.

## 2. Materials and Methods

Bsal growth conditions: The Bsal-type strain (AMFP 13/01), originating from a Bsal-positive fire salamander (*Salamandra salamandra*) [2], was cultured in tryptone-gelatin hydrolysate-lactose (TGhL) broth and incubated in 75 cm^2^ cell culture flasks at 15 °C for 5–7 days to promote growth. We collected Bsal spores from a full-grown culture containing mature sporangia. Upon release of the zoospores, the culture medium was collected and filtered through a sterile mesh filter with a pore size of 10 μm (PluriSelect, Leipzig, Germany). The resulting flow-through, enriched with zoospores at >90% purity, was used as the zoospore fraction for further experimentation. Zoospore viability and motility was confirmed using light microscopy.

Cell culture media: Three distinct media compositions were employed in this study. For the routine growth and maintenance of A6 cells, complete growth medium was used, comprising 74% NCTC 109 medium, 15% distilled water, 10% fetal bovine serum (FBS), and 1% of a 10,000 U/mL penicillin-streptomycin solution (all from Gibco, part of Thermo Fisher Scientific, Waltham, MA, USA). Upon exposure of A6 cells to Bsal, L-15-based cell medium was employed (Gibco). To facilitate the motility of Bsal zoospores during the initial contact phase, infection medium 30% was used, comprising 30% L-15 medium, 60% distilled water, and 10% FBS. Following the initial 2 h period, during which Bsal zoospores actively migrated towards the A6 cells, the medium was replaced with infection medium 70%, consisting of 70% L-15 medium, 20% distilled water, and 10% FBS. This strategic modulation of media composition was implemented to preserve Bsal zoospore motility during the critical initial attachment phase and to promote subsequent fungal development, while simultaneously maintaining the health and viability of the A6 host cells throughout the infection process.

Cell culture: The *Xenopus laevis* kidney epithelial cell line A6 (CCL 102, ATCC, Manassas, VA, USA) was cultured in 75 cm^2^ cell culture flasks and maintained in complete growth medium [19]. Cells were incubated at 26 °C with 5% CO_2_ until reaching confluence. Upon confluence, cells were detached using trypsin, washed with 70% Hanks’ Balanced Salt Solution with Ca^2+^ and Mg^2+^ (HBSS+) (Gibco) by centrifugation for 5 min at 1500 rpm, and then resuspended in the appropriate cell culture medium for invasion assays.

Toxicity of cytochalasin D on Bsal zoospores and A6 cells: To examine the role of cytoskeletal changes in A6 cells during Bsal infection, cytochalasin D (CD), an actin polymerization inhibitor, was used to investigate the impact of disrupting actin dynamics. A 24 h exposure to CD was chosen to encompass the initial invasion step of Bsal infection, allowing us to assess the impact of actin disruption during the early stages of infection. Since it is important to use concentrations that do not harm the cells or Bsal, the toxicity of CD was first evaluated.

To determine the impact of CD (Sigma Aldrich (St. Louis, MO, USA): C2618) on Bsal viability, 7.5 × 10^5^ zoospores per well were seeded in TGhL medium containing varying concentrations of CD in 24-well plates. Following 24 h of exposure, the medium was replaced with 1 mL of control TGhL medium, and the cultures were further incubated for 4 days at 15 °C. On day 5 post infection (p.i.), Bsal was harvested from the bottom of the well using a 1 mL pipette tip, combined with the supernatant, and centrifuged for 5 min at 3000 rpm. The resulting pellet was resuspended in 50 μL of Prepman Ultra reagent (Thermo Fisher Scientific, Waltham, MA, USA), heated for 10 min at 100 °C, and allowed to cool to room temperature for 2 min. Subsequently, the tubes were centrifuged for 2 min at 13,000 rpm, and total Bsal counts were determined using quantitative PCR [20]. All treatments were tested in at least three independent experiments (biological replicates) with a minimum of three technical replicates.

To determine the EC_50_ of CD on A6 cells, we conducted a neutral red assay as per previously established protocols [21]. Briefly, A6 cells were seeded in 96-well microplates at a density of approximately 2 × 10^5^ cells per well in growth medium and allowed to attach overnight at 26 °C with 5% CO_2_. Subsequently, the cells were exposed to varying concentrations of CD in 70% infection medium for 24 h. Negative controls comprised cells treated with 70% infection medium, while positive controls were treated with 1% Triton X-100. Vehicle controls were treated with DMSO (0.02%), matching the highest DMSO concentration present in the experimental conditions. Cytotoxicity was assessed by adding 200 μL of freshly prepared neutral red solution (33 μg/mL in 70% L-15 medium without phenol red), prewarmed to 26 °C, to each well, followed by incubation at 26 °C for an additional 2 h. After washing the cells twice with 70% HBSS+, 200 μL of extraction solution (ethanol/Milli-Q water/acetic acid, 50/49/1, *v*/*v*/*v*) was added to each well and the plate was shaken for 10 min. Absorbance was measured at 540 nm using a microplate ELISA reader (Multiskan Go, Thermo Scientific, Merelbeke, Belgium). Relative viability, expressed as a percentage compared to the negative control (considered as 100%), was calculated using the formula 100 × ((a − b)/(c − b)), where a = OD_540_ from wells treated with CD, b = OD_540_ from blank wells, and c = OD_540_ from untreated control wells. All treatments were tested in at least three independent experiments (biological replicates) with minimally three technical replicates. The EC_50_ of CD for A6 cells was calculated by fitting a dose–response curve using GraphPad Prism software 8.4.3.

Fluorescent RCA staining to examine the expression of β-galactose in A6 cells: Since β-galactose, presented on the cell surface, is involved in Bsal attachment to the host cell [18], we assessed the expression of β-galactose in A6 cells. Cells were seeded at a density of 5 × 10^4^ cells per well in 24-well tissue culture plates containing collagen-coated glass coverslips and allowed to adhere overnight at 26 °C with 5% CO_2_. Following attachment, the cells underwent three washes with 70 HBSS+ and were subsequently fixed for 10 min with 3.7% paraformaldehyde in 70% HBSS+. We treated the cells with 0.1% Triton X-100 for 2 min to permeabilize them and compared this condition to cells that were not permeabilized. The cells were then treated with fluorescein-labeled RCA I (Ricinus communis agglutinin I) (Vector Laboratories, Newark, CA, USA) at a concentration of 25 µg/mL for 30 min. RCA I was diluted in lectin binding buffer (10 mM Hepes, 0.15 M NaCl, pH 7.5). To serve as a negative control, RCA I was pre-incubated with 200 mM galactose before application to the cells, thereby inhibiting lectin binding. A positive control consisting of a slide of fire salamander ventral skin was included, sourced from prior infection trials [18]. Following three washes with 70% HBSS+, the samples were mounted using ProLong™ Gold antifade mountant (Thermo Fisher Scientific), and subjected to analysis using fluorescence microscopy.

In vitro model to assess invasion and intracellular maturation of Bsal in A6 cells: The experimental protocol was adapted from Verbrugghe et al. (2019) [15]. To evaluate the interactions between Bsal and A6 cells, 10^5^ A6 cells were seeded per well in 24-well tissue culture plates containing collagen-coated glass coverslips and allowed to attach overnight at 26 °C with 5% CO_2_. Following attachment, the cells were washed three times with 70% HBSS+ and subsequently inoculated with Bsal zoospores in infection medium (30%) at a multiplicity of infection (MOI) of 1:10. The cells were then incubated for 2 h at 15 °C with 5% CO_2_. After incubation, non-attached spores were removed by gently washing the cells three times with 70% HBSS+. The cells were then supplemented with infection medium (70%) and incubated for 1 to 8 days p.i., with the cell medium replenished every 3 days. Cell viability 7 days p. i. was assessed using neutral red [21].

Visualization of Bsal-A6 cell interactions was performed as follows: infected cells were stained with 9 μM CellTracker^TM^ Green CMFDA (Thermo Fisher Scientific) for 45 min at 15 °C with 5% CO_2_, followed by gentle washing with 70% HBSS+. Extracellular Bsal was visualized by incubating the infected cells with Calcofluor White stain (50 μg/mL in 70% HBSS+, Sigma Aldrich) for 10 min, followed by two washes with HBSS+. The cells were then fixed for 10 min with 3.7% paraformaldehyde in 70% HBSS+, permeabilized for 2 min with 0.1% Triton X-100, and incubated for 60 min with a polyclonal antibody against Bsal (type strain AMFP13/1) produced in rabbit [22]. After three washes with 70% HBSS+, the samples were incubated with a monoclonal goat anti-rabbit Alexa Fluor 568 antibody (1/750 dilution, Thermo Fisher Scientific) to visualize both intracellular and extracellular Bsal. Following a 1 h incubation, the samples were washed three times with 70% HBSS+, mounted using ProLong™ Gold antifade mountant, and analyzed using fluorescence microscopy. Sham-infected cells were included as a negative control.

To determine the proportion of intracellular (endobiotic) and extracellular (epibiotic) growth of Bsal 3 days p. i., three biological replicates were performed, each with at least three technical replicates and two imaged regions per replicate. ImageJ software version 1.54g was used for image analysis, applying a threshold to each channel followed by particle analysis to quantify the total area.

Fluorescent actin staining to assess the cytoskeletal changes during Bsal infection of A6 cells: To visualize cytoskeletal rearrangements during Bsal infection, A6 cells were subjected to a fluorescent staining using Phalloidin Texas Red^®^-X (Thermo Fisher Scientific), a high-affinity probe targeting F-actin labeled with the red fluorescent dye Texas Red^®^-X. Following infection of A6 cells as detailed in the “in vitro model” section, extracellular Bsal was detected by incubating the infected cells with Calcofluor White stain (50 μg/mL in 70% HBSS+) for 10 min, followed by two washes with HBSS+. Subsequent steps involved fixation of the cells for 10 min with 3.7% paraformaldehyde in 70% HBSS+, permeabilization for 2 min with 0.1% Triton X-100, and incubation with a polyclonal antibody against Bsal for 60 min [22]. After three washes with 70% HBSS+, the samples were incubated with a monoclonal goat anti-rabbit Alexa Fluor 488 antibody (1/250, Thermo Fisher Scientific) to visualize both intracellular and extracellular Bsal. Following a 1 h incubation, the cells were stained with Phalloidin Texas Red^®^-X (165 nM) in 70% HBSS+ and incubated for 1 h. Finally, the cells were washed three times with 70% HBSS+, mounted using ProLong™ Gold antifade mountant, and analyzed using fluorescence microscopy. Sham-infected cells served as a negative control.

TUNEL and caspase staining of tissue slides to assess induction of apoptosis: The TUNEL assay is commonly used to investigate apoptotic cells by measuring DNA damage. However, since DNA damage can also occur in other cell death mechanisms, such as necrosis, a control caspase-3 staining was performed. Skin tissues from Bsal-infected fire salamanders, sourced from prior infection trials, were utilized to investigate the occurrence of apoptosis during Bsal infection [23]. The skin sections, fixed in 4% phosphate-buffered formaldehyde, underwent de-paraffinization in xylene followed by hydration through a series of alcohol gradients. Subsequently, the tissue slides were subjected to antigen retrieval in 0.1 M citrate buffer (pH 6.0) using microwave irradiation. After cooling, a serum blocking step with 10% goat serum was conducted for 30 min, followed by washing with PBS for 10 min. For the TUNEL staining, the samples were then incubated with polyclonal antibodies against Bsal (1/1000, 60 min) [22]. Following three washes, the tissues were exposed to a TUNEL reaction mixture according to the manufacturer’s instructions (In Situ Cell Death Detection Kit, TMR red, Sigma Aldrich), along with a secondary antibody (goat anti-rabbit Alexa 488, 1/250). After an additional hour of incubation, the sections were washed and incubated with Hoechst stain (10 µg/mL, Sigma Aldrich). For the caspase-3 staining, both the polyclonal antibodies against Bsal and the anti-caspase 3 antibody (C8487, Sigma Aldrich) are produced in rabbit. Therefore, different tissue slides were treated with the chytrid antibodies (1/1000, 60 min) or the anti-caspase 3 antibody (1/1000, 90 min). Following washing steps, slides targeting chytrid cells were incubated with a secondary antibody goat anti-rabbit Alexa 488 (1/250), while those treated with the antibody against caspase 3 received goat anti-rabbit Alexa 568 (1/500). Subsequently, sections were incubated with Hoechst stain (10 µg/mL). Finally, all sections were mounted using ProLong™ Gold antifade mountant. Positive controls included tissues treated with DNase I recombinant (3000 U/mL) for 10 min.

Ex vivo *Bsal* infection of skin explants: We used tissue slides derived from a prior ex vivo experiment [24], in which Bsal was introduced to the skin of fire salamanders for 2–24 h. In short, tail clips of fire salamanders were collected, rinsed thoroughly with HPLC H_2_O, and they were placed in 24-well plates together with 1.5 mL of Bsal zoospore suspension (2 × 10^7^ zoospores/mL in distilled H_2_O). Samples were incubated at 15 °C, 5% CO_2_ for 2, 4, or 24 h, after which they were washed three times with HPLC H_2_O to remove non-adherent zoospores. Subsequently, the tail clips were fixed in 4% phosphate buffered formaldehyde, embedded in paraffin, and 5 µm sections were cut and stained with Gomori methenamine silver stain (GMS), or they were subjected to a fluorescent staining. Therefore, tissue sections were de-paraffinized in xylene and hydrated through a graded series of alcohols. Subsequently, the tissue slides were subjected to antigen retrieval in 0.1 M citrate buffer (pH 6.0) using microwave irradiation. After cooling, a serum-blocking step with 10% goat serum was conducted for 30 min, followed by washing with PBS for 10 min. The samples were then incubated with the chytrid antibodies (1/1000, 60 min) [22]. After washing three times for 5 min, the tissues were incubated with a secondary antibody goat anti-rabbit Alexa 488 (1/250). After an incubation of 1 h, the samples were washed with 70% HBSS+ and incubated with Texas Red-X phalloidin (165 nM). After 1 h, the samples were washed 3 times and incubated with Hoechst stain (10 µg/mL). Finally, the sections were mounted using ProLong™ Gold antifade mountant.

## 3. Results

### 3.1. A6 Cells Express Beta-Galactose and Can Be Used to Mimic the Bsal Life Cycle In Vitro

RCA I staining of A6 cells demonstrates the distribution of β-galactose, notably concentrated on the cell surface (Figure 1a), with presence extending into the cytoplasmic and nuclear compartments (Figure 1b), suggesting their potential as a suitable model system to investigate Bsal interactions with host cells. Following inoculation of Bsal in the A6 cells, we observed the formation and growth of germ tubes within 24 h, facilitating the active penetration of host cells (Figure 1c,d). Subsequently, the intracellular life cycle of Bsal commenced as endobiotic growth, characterized by the development of thalli within the host cell. By day 3 p.i., the formation of both monocentric, in which only one zoosporangium forms, and colonial thalli, in which multiple sporangia form along internal septa, was evident (Figure 1e). Further maturation of the thalli led to the presence of large sporangia within the A6 cells, with subsequent intracellular release of new spores (Figure 1f,g). Subsequently, observations of numerous heavily infected cells and areas devoid of cells from day 7 p.i. onward suggested widespread host cell death and release of intracellular contents (Figure 1h and Appendix A), paralleling in vivo infection scenarios where cell death is observed around Bsal lesions that span all epidermal cell layers (Figure 1i and Appendix A). In addition to endobiotic growth, we also observed epibiotic growth, characterized by Bsal development outside the host cells. Following germ tube protrusion into the cells, Bsal demonstrates the capacity for extracellular growth, maturing into mature sporangia without the need for intracellular transfer of its contents (Figure 1j,k). At day 3 p.i., the proportion of extracellular Bsal was 49.99% ± 8.91 (SEM) across all biological and technical replicates.

### 3.2. Bsal Host Cell Entry: Active Penetration Independent of Host Actin Dynamics

To investigate the invasion dynamics of Bsal and A6 cells, with a focus on the actin cytoskeleton, we administered CD during the first 24 h p.i.—a timeframe coinciding with the expected onset of invasion. Due to variability in concentration effects and the potential for acute toxicity in both Bsal and A6 cells, toxicity assessments were performed to establish a concentration range that ensures the viability of both the cells and the pathogen. The EC_50_ value of CD for A6 cells was established at 241.5 nM (Appendix A), a concentration that did not affect Bsal growth (Appendix A). Therefore, we examined the dynamic progression of Bsal infection within A6 cells, focusing on the modulation of the host cell cytoskeleton, by administering 200 nM CD during the early stages of Bsal–host interaction.

Within 24 h of inoculation, Bsal demonstrated efficient intracellular penetration and the concurrent induction of pore formation within the host cytoskeleton, facilitating the transfer of its contents (Figure 2a–a″). Treatment with CD failed to impede this process, suggesting a mode of entry independent of conventional host actin dynamics like endocytosis (Figure 2b–b″). At the 3-day time point, subsequent observations revealed that Bsal thalli underwent intracellular maturation, which was accompanied by a marked increase in actin assembly around the pathogen (Figure 2c,d). By day 8 p.i., the liberation of intracellular contents coincided with the dissolution of host cell actin structures, indicative of potential mechanisms underlying host cell damage and pathogen dissemination (Figure 2e,f). The absence of positive staining with Calcofluor White in the zoospores indicates a lack of chitin in their cell walls (Figure 2e). This observation suggests that these zoospores are freshly released, as chitin deposition typically occurs during encystation [25]. Notably, inhibition of actin rearrangements during the initial 24 h infection window failed to abrogate Bsal invasion, resulting in observations akin to those of untreated cells (Figure 2g,h).

The majority of microscopic analyses conducted on skin tissue from Bsal-infected animals have primarily focused on specimens obtained during the end stages of infection, characterized by the presence of visible lesions. Consequently, direct evidence of germ tube formation and active penetration remains elusive. We therefore focused on tissue slides from prior ex-vivo experiments [24], where Bsal was introduced to the skin of fire salamanders and visualized shortly after, allowing us to observe early host–pathogen interactions. Two hours following inoculation, Bsal attaches firmly to the stratum corneum (Figure 3a). This initial phase is succeeded by a discernible thickening of the cellular wall by the 4 h time point, marking the onset of encystment (Figure 3b). At 24 h p.i., active penetration of the stratum corneum is observed. This invasive mechanism is typified by germ tube-mediated invasion (Figure 3c,d), facilitating the subsequent transfer of intracellular contents into the host cells (Figure 3e).

## 4. Discussion

In this study, we successfully characterized the life cycle of Bsal within A6 cells, identifying key stages such as attachment to beta-galactosides, encystment, intracellular content transfer, intracellular maturation, and subsequent Bsal release. These stages closely align with both in vivo and ex vivo findings. Once Bsal invades A6 cells, it actively transfers its contents intracellularly, leading to the maturation of monocentric and predominantly colonial thalli. This process mirrors in vivo observations, where *Bsal* primarily forms colonial thalli with multiple-walled sporangia in the amphibian skin [2]. The maturation of colonial thalli within host cells is a critical aspect of Bsal pathogenicity, as these thalli act as reservoirs for further sporangial development and pathogen dispersal.

Our findings further demonstrate that the culmination of the Bsal infection cycle results in apoptosis of epidermal cells, which is evident around Bsal-induced lesions in infected amphibians. This apoptotic cell death underscores the destructive impact of Bsal on host tissue integrity. Similarly, in our in vitro system, infected A6 cells undergo cell death, with the entire infection cycle from Bsal attachment to release spanning at least 7 days, consistent with the in vivo situation [2]. This consistency suggests that our in vitro model effectively replicates the timeline and key stages of Bsal development observed in vivo.

Although our in vitro model closely replicates the in vivo situation, several differences have been observed. One notable difference is the formation of an “actin cocoon” around Bsal thalli by the third day p.i. This phenomenon, commonly observed with various intracellular pathogens to maintain niche integrity and promote cytosolic escape [26], has not been documented for Bsal or Bd in amphibian skin, where the thalli typically reside freely in the cytosol. The formation of the actin cocoon in vitro may be influenced by the use of A6 cells, which originate from non-keratinocyte sources. Another key difference in comparison to host skin is the occurrence of epibiotic growth, in which Bsal, similar to Bd [15,17], develops outside the host cells and utilizes them as a nutrient source. While this saprotrophic growth mechanism has not been described in the amphibian skin, it has been documented on grass plant materials, where it plays a significant role in environmental survival and persistence [27].

Bsal zoospores commonly undergo germ tube formation [2], suggesting a mechanism involving germ tube-mediated invasion. However, detailed characterization of the early infection dynamics has remained limited. In this study, we used the in vitro model to demonstrate that Bsal primarily invades host cells through germ tube-mediated penetration rather than endocytosis—a finding supported by both our in vitro and ex vivo investigations. Following attachment and encystment, Bsal zoospores form germ tubes that penetrate the stratum corneum within 24 h, allowing for direct transfer of intracellular contents. This active penetration mechanism aligns with the expanded presence of protease-encoding genes in Bsal, which are hypothesized to play a critical role in the early stages of zoospore colonization and entry into host cells [4,27]. Although the precise function of these proteases remains to be elucidated, it is likely that they facilitate enzymatic digestion of the epithelial surface, thereby enhancing mechanical entry into host cells.

A key aspect to consider in our study is the use of a single cell line, particularly one not derived from amphibian keratinocytes. To our knowledge, only six caudatan cell lines are available in repositories worldwide, and they are all derived from embryonic or whole-larval tissues [28]. While anuran-derived epithelial-like cell lines, such as Xela DS2 and Xela VS2 [29], could offer valuable models for Bd research, the lack of a keratinocyte-derived caudatan cell line remains a major challenge in Bsal research, and developing such a model would greatly enhance physiological relevance.

Bsal susceptibility varies widely among amphibians, from hypersusceptible species such as fire salamanders, which fail to mount a protective response and rapidly succumb to infection, to species exhibiting variable responses [30]. Host susceptibility has been linked to skin glycosylation patterns, with cutaneous galactose content correlating with Bsal colonization intensity [18]. Our in vitro model closely reflects the rapid and severe disease progression observed in hypersusceptible species, where Bsal proliferates unchecked, leading to host cell destruction within a timeframe consistent with in vivo observations in fire salamanders. Based on our own data extending up to 14 days post-infection, prolonging the observation period only reinforces this trajectory, with no indication of persistence or recovery. Studying prolonged infection dynamics, including persistence or potential host recovery, would possibly require alternative cell models. Future efforts to establish keratinocyte-derived caudatan cell lines, particularly those with differing β-galactose content, could provide valuable insights into species-specific differences in susceptibility and could allow for a more nuanced investigation of chronic infections.

However, colonization alone does not necessarily predict disease outcome, as innate and acquired immune responses and environmental factors ultimately shape infection progression [18]. Amphibian immune mechanisms are critical to understanding Bsal infection. The amphibian skin plays a central role in mucosal immunity, with the microbiome and antimicrobial peptides providing protection against pathogens, including Bsal [6,12]. In terms of innate and adaptive immune responses, Bsal has been shown to suppress these defenses in infected salamanders, impairing their ability to mount an effective response, suggesting immune-dampening properties [31]. As our study focused primarily on Bsal–host cell interactions at the cellular level, it did not account for the role of host immunity or pathogen adaptation. Incorporating immune components, such as conditioned media containing antimicrobial peptides or even co-culture systems with immune cells, could provide a more comprehensive picture of host–pathogen interactions in future studies. Furthermore, while the controlled nature of in vitro systems allows for precise dissection of specific host–pathogen interactions, it does not replicate the fluctuating environmental conditions that influence Bsal infectivity and survival in natural settings. Temperature, humidity, salinity, and even substrate are environmental factors known to influence amphibian disease outcomes [7,32,33,34]. Future applications of this model could involve controlled environmental manipulations to assess how these factors influence Bsal–host interactions, bridging the gap between laboratory-based findings and ecological relevance.

In this study, we focused on early host–pathogen interactions using tissue slides from fire salamanders, based on previous in vivo and ex vivo work [18,23,24]. However, due to inherent biological differences across models, direct quantitative comparisons of infection rates or severity scores between in vivo, ex vivo, and in vitro systems are challenging. Factors such as species variability in response to Bsal, immune differences, environmental conditions, and infection dose all contribute to the complexity of infection progression and severity [6,7,30]. While in vivo and ex vivo models can be quantitatively assessed using qPCR, this approach is complicated in our in vitro system due to the interference of epibiotic growth, which confounds accurate fungal load measurement. Numerous pathogens, such as *Salmonella*, benefit from established in vitro assays like the gentamicin protection assay, which enables accurate quantification of intracellular bacteria [35]. Optimizing a similar tool with our in vitro model could facilitate direct intracellular quantification of Bsal using qPCR or EMA-qPCR, improving our ability to assess fungal load more precisely [36].

Beyond environmental factors or immune components, this model could also be used to investigate other key aspects of Bsal infection. Comparative studies using different Bsal isolates could provide insights into strain virulence, similar to research conducted on Bd in Greener et al. (2020) [16]. Additionally, it could be used to evaluate antifungal treatments and environmental control measures (e.g., temperature, pH, or disinfectants) to identify factors that influence Bsal growth and host–pathogen interactions [37]. Moreover, the model can be leveraged to explore host–pathogen interactions in greater detail, such as examining the effects on host cell and pathogen gene expression [38].

In conclusion, this study represents a significant advancement in our understanding of Bsal infection dynamics by establishing an in vitro infection model utilizing A6 cells, which closely mimics in vivo conditions. This model allowed us to investigate early interactions between Bsal and host cells, with a specific focus on actin rearrangements during the invasion process. Our findings not only highlight the adaptability and versatility of this experimental system but also provide a valuable platform for future research aimed at further elucidating the pathogenesis of Bsal.

## Figures and Tables

**Figure 1 jof-11-00156-f001:**
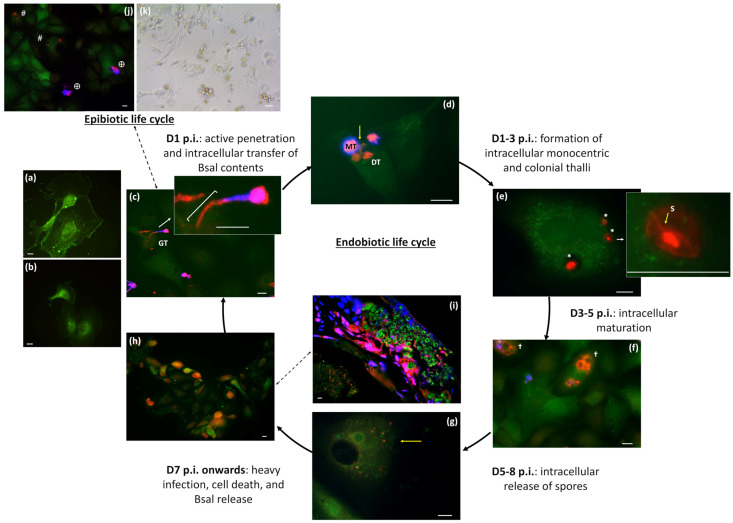
Bsal in vitro infection cycle. Visualization of terminal β-galactosides in (**a**) non-permeabilized and (**b**) permeabilized A6 cells using RCA staining. (**c**–**h**,**j**) Overlay images of fluorescently labeled Bsal-infected A6 cells (green cell tracker), extracellular Bsal (Calcofluor White, blue), and extra- and intracellular Bsal (Alexa Fluor 568, red) at various time points p.i. (**i**) Fluorescent TUNEL staining of cell death (red) in skin tissue of a heavily Bsal-infected fire salamander (Alexa Fluor 488, green). Nuclei were stained with Hoechst (blue). (**c**) Within 24 h of inoculation, germ tubes (GTs) are formed, penetrating A6 cells (bracket). (**d**) Active penetration of Bsal results in intracellular transfer (yellow arrow) of the content from mother thalli (MT) to new daughter thalli (DT). (**e**) Formation of monocentric and colonial thalli (*) with internal septa (S) observed at day 1–3 p.i. (**f**) At day 3–5 p.i., intracellular thalli mature into sporangia (†), intracellularly releasing the spores at day 5–8 p.i. (yellow arrow) (**g**). (**h**) From day 7 p.i. onwards, heavily infected cells and large areas nearly devoid of cells indicate cell death and release of intracellular contents, which mirrors the in vivo situation (**i**), where similar observations of heavy infection and cell death occur in the skin of fire salamanders. Clear cell death is observed around intracellular thalli, spanning all epidermal cell layers. (**j**) In addition to the endobiotic life cycle (#), epibiotic growth of Bsal (⊕) is also observed on day 3 p.i. (**k**) Light microscopy image of Bsal-infected A6 cells at 8 days p.i., depicting extracellular growth of Bsal. Scale bar = 10 µm.

**Figure 2 jof-11-00156-f002:**
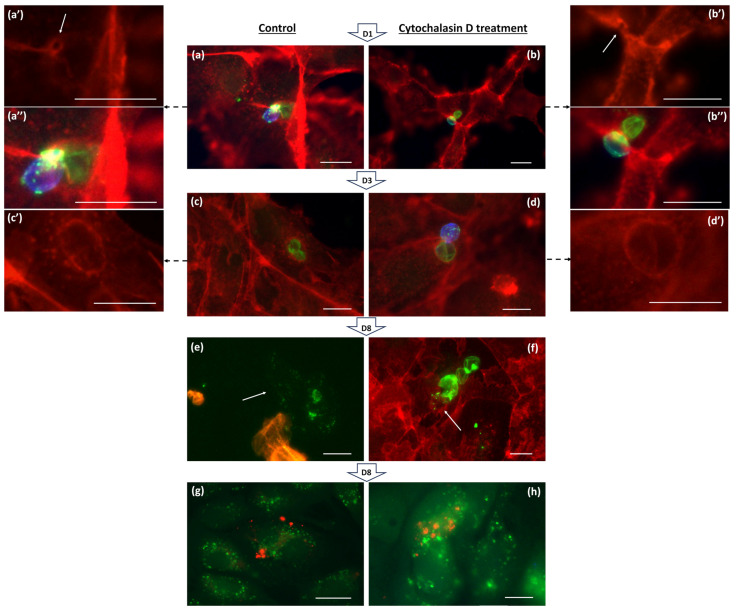
Direct entry of Bsal into host cells. Fluorescent overlay images depict Bsal-infected A6 cells at various time points p.i. Panels (**a**–**f**) show the cytoskeleton of A6 cells stained with Phalloidin Texas Red (red), extra- and intracellular Bsal labeled with Alexa Fluor 488 (green), and extracellular Bsal stained with Calcofluor White (blue). Panels (**g**,**h**) display Bsal-infected A6 cells (green cell tracker), extra- and intracellular Bsal labeled with Alexa Fluor 568 (red), and extracellular Bsal stained with Calcofluor White (blue). Host–pathogen interactions were analyzed (**b**,**d**,**f**,**h**) with and (**a**,**c**,**e**,**g**) without inhibition of cytoskeletal rearrangements during the initial 24 h. Details are provided in the left and right panels, with (**a′**–**d′**) showing only the red signal and (**a″**–**b″**) showing the overlay details. Within 24 h after inoculation, Bsal penetrates the host cell, transferring its contents intracellularly and inducing pore formation in the host cytoskeleton (**a****′**: white arrow). (**b**–**b″**) Treatment with CD during the initial 24 h does not inhibit this process. (**c**,**d**) At 3 days p.i., intracellular maturation of thalli is observed, accompanied by actin assembly around the pathogen (**c****′**–**d****′**). (**e**–**f**) By day 8 p.i., release of intracellular contents occurs (white arrows), coinciding with loss of actin structures. (**h**) Inhibition of actin rearrangements during the first 24 h of infection does not prevent Bsal from invading host cells, resulting in similar observations to untreated cells (**g**). Scale bar = 10 µm.

**Figure 3 jof-11-00156-f003:**
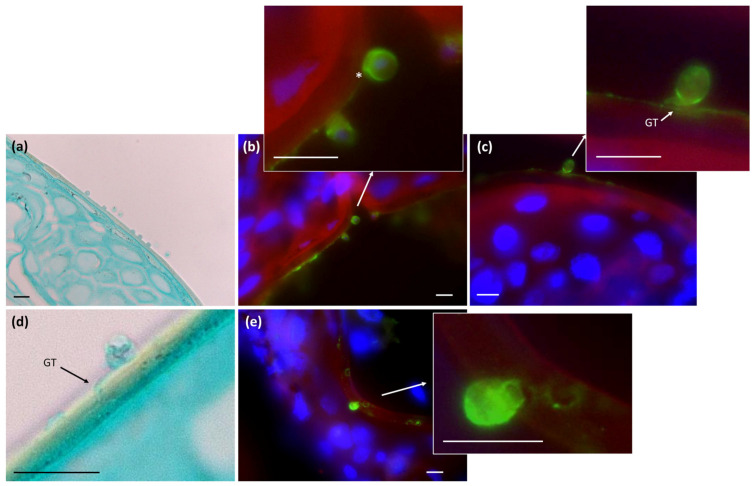
Temporal dynamics in early Bsal host interactions. Composite image depicting early interactions of Bsal with the stratum corneum of FTE explants from fire salamanders that were exposed to Bsal zoospores for (**a**) 2, (**b**) 4, or (**c**–**e**) 24 h [24], using (**a**,**d**) GMS and (**b**,**c**,**e**) fluorescent stainings. (**a**) Two hours after inoculation, attachment to the stratum corneum is evident. (**b**) Subsequently, at the 4 h time point, distinct cell wall thickening (*) indicates encystment of Bsal zoospores. By 24 h p.i., there is evident active penetration of the stratum corneum, typified by (**c**,**d**) germ tube (GT)-mediated invasion and (**e**) intracellular content transfer. Scale bar = 10 µM. For a detailed overview of the skin morphology and key structures, see Appendix A.

## Data Availability

The original data presented in the study are openly available in FigShare at https://doi.org/10.6084/m9.figshare.27239985.

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
