# Peer review of "In Vitro Infection Model Using A6 Cells Sets the Stage for Host–*Batrachochytrium salamandrivorans* Exploration"

_jof, 2025, doi:10.3390/jof11020156_

Round 1

Reviewer 1 Report

This study by Verbrugghe et al. employs multiple complementary methods, including cell culture, fluorescence microscopy, qPCR, and biochemical assays, to examine different aspects of Bsal infection. It successfully establishes an in vitro infection model using A6 cells, attempting to mimic in vivo conditions, making it a valuable tool for studying early-stage Bsal infection dynamics. Verbrugghe et al. also demonstrate that Bsal primarily invades host cells via germ tube-mediated penetration rather than endocytosis.

However, there are several limitations to this study: 

The study focuses on a single cell line (A6 kidney epithelial cells), which may not fully represent other amphibian tissues susceptible to Bsal infection. A6 cells originate from non-keratinocyte sources, which could influence host-pathogen interactions differently than amphibian epidermal cells.

The experiments are conducted under controlled lab conditions, which may not fully capture the environmental complexity affecting Bsal behavior in the wild. Factors such as temperature fluctuations and other environmental constraints were not considered in this study.

Amphibians have complex immune responses, including antimicrobial peptides and inflammatory reactions, which cannot be replicated in an in vitro system. While the study focuses on host cell interactions, it does not account for how immune factors might influence Bsal infection, progression, or clearance. Addressing this in the discussion could strengthen the study’s relevance.

The study tracks Bsal development over only seven days, potentially missing later-stage processes or chronic infection outcomes. A longer observation period might provide more insights into the pathogen’s long-term behavior.

The current model does not account for potential host-specific variations, such as differences in susceptibility among various amphibian species. Expanding the study to include multiple amphibian cell lines could enhance its applicability.

Figure 3 (lines 332-339), for a reader who is not familar with the morphology will find this hard to understand. Maybe an illustration, highlighting the cell morphology is required. The abbreviation "GT" is missing in the caption.

Author Response

For research article

Response to Reviewer Comment

Point-by-point response to Comments and Suggestions for Authors

Comment 1: The study focuses on a single cell line (A6 kidney epithelial cells), which may not fully represent other amphibian tissues susceptible to Bsal infection. A6 cells originate from non-keratinocyte sources, which could influence host-pathogen interactions differently than amphibian epidermal cells.

Response 1: Use of a single cell line: While our study utilizes a single cell line that is not derived from amphibian keratinocytes, A6 cells provide a well-characterized and practical system for demonstrating the feasibility of an in vitro model for Bsal infection. Importantly, A6 cells express β-galactose, a key receptor for Bsal attachment, making them a relevant model for studying early invasion dynamics. To further strengthen the translational value of our  findings, we agree that future studies should validate the infection model in keratinocyte-derived cell models or explant cultures to confirm their applicability to salamander skin tissues.

With this study, we aim to demonstrate that in vitro modeling of Bsal infection is feasible, similar to existing in vitro models for Bd, which have also been successfully applied in other cell line models, including Xenopus fibroblasts (Webb et al., 2024). The recently described Xela DS2 and Xela VS2 skin epithelial-like cell lines could represent a promising addition to Bd in vitro research (Bui-Marinos et al., 2020), given their epithelial-like characteristics and skin-derived origin. However, since both A6 and Xela cell lines originate from non-caudatan species, the advantages of switching to an alternative anuran-derived epithelial cell line remain limited in the context of Bsal research.

For Bsal, the challenge is more complex. A meaningful step forward would require an epithelial cell line derived from a Bsal-susceptible caudatan species, ideally a keratinocyte-based model. However, such a cell line is currently unavailable. To our knowledge, only six caudatan cell lines exist in repositories worldwide (RCB), all originating from two species (Hynobius nebulosus and Hynobius tokyoensis). These lines were established from embryos or whole larvae primary cultures of unknown tissue type rather than differentiated epithelial tissues (Douglas et al., 2023). Currently, none of the commercially available caudatan cell lines are derived from keratinocytes or skin epithelium, highlighting the challenge of developing a truly representative in vitro model for Bsal pathogenesis. Our ongoing efforts focus on isolating primary keratinocytes and developing an epithelial-like cell line from Bsal-relevant salamander species. However, unlike mammalian keratinocytes, establishing stable caudatan keratinocyte cultures presents significant technical challenges. Addressing this gap will be essential for advancing in vitro studies of Bsal infections in physiologically relevant cell models, and this is an area we are actively working on.

We appreciate the reviewer's thoughtful remark and have addressed this point in the manuscript (Lines 405-411): A key aspect to consider in our study is the use of a single cell line, particularly one not derived from amphibian keratinocytes. To our knowledge, only six caudatan cell lines are available in repositories worldwide, and they are all derived from embryonic or whole-larval tissues [28]. While anuran-derived epithelial-like cell lines, such as Xela DS2 and Xela VS2 [29], could offer valuable models for Bd research, the lack of a keratinocyte-derived caudatan cell line remains a major challenge in Bsal research, and developing such a model would greatly enhance physiological relevance.

Comment 2: The experiments are conducted under controlled lab conditions, which may not fully capture the environmental complexity affecting Bsal behavior in the wild. Factors such as temperature fluctuations and other environmental constraints were not considered in this study.

Response 2: Controlled lab conditions vs. environmental complexity: We agree that our study does not account for the full range of environmental factors that influence Bsal behavior in the wild, such as temperature fluctuations, humidity, and substrate composition. However, the controlled nature of in vitro systems allows for precise dissection of specific host-pathogen interactions, which can serve as a foundation for subsequent studies under more complex conditions.

We have addressed this point in the manuscript (Lines 442-449): Furthermore, while the controlled nature of in vitro systems allows for precise dissection of specific host-pathogen interactions, it does not replicate the fluctuating environmental conditions that influence Bsal infectivity and survival in natural settings. Temperature, humidity, salinity, and even substrate are environmental factors known to influence amphibian disease outcomes [7,32-34]. Future applications of this model could involve controlled environmental manipulations to assess how these factors influence Bsal-host interactions, bridging the gap between laboratory-based findings and ecological relevance.

Comment 3: Amphibians have complex immune responses, including antimicrobial peptides and inflammatory reactions, which cannot be replicated in an in vitro system. While the study focuses on host cell interactions, it does not account for how immune factors might influence Bsal infection, progression, or clearance.

Response 3: Lack of amphibian immune components: As the reviewer notes, amphibians possess intricate immune defenses, including antimicrobial peptides, mucosal immunity, and inflammatory responses. These complex immune mechanisms are not fully replicated in our in vitro model, which primarily focuses on Bsal-host cell interactions independent of immune modulation.

This was addressed in the discussion at lines 430-442: However, colonization alone does not necessarily predict disease outcome, as innate and acquired immune responses and environmental factors ultimately shape infection progression [18]. Amphibian immune mechanisms are critical to understanding Bsal infection. The amphibian skin plays a central role in mucosal immunity, with the microbiome and antimicrobial peptides providing protection against pathogens, including Bsal [6,12]. In terms of innate and adaptive immune responses, Bsal has been shown to suppress these defenses in infected salamanders, impairing their ability to mount an effective response, suggesting immune-dampening properties [31]. As our study focused primarily on Bsal-host cell interactions at the cellular level, it did not account for the role of host immunity or pathogen adaptation. Incorporating immune components, such as conditioned media containing antimicrobial peptides or even co-culture systems with immune cells, could provide a more comprehensive picture of host-pathogen interactions in future studies.

Comment 4: The study tracks Bsal development over only seven days, potentially missing later-stage processes or chronic infection outcomes. A longer observation period might provide more insights into the pathogen’s long-term behavior.

Response 4: When cultivating Bsal in TghL medium under controlled laboratory conditions, a generation time of five days at 15°C is observed (Martel et al., 2013). However, within our in vitro model, this cycle was prolonged to at least 7 days. The growth dynamics of Bsal in vivo are influenced by various factors including the host species, inoculation dose, Bsal strain, and ambient temperature (Stegen et al., 2017). Consequently, conventional TGhL assays may potentially overestimate Bsal proliferation when compared to cellular and in vivo infection models. While we acknowledge that a longer observation period may provide insights into chronic infection outcomes, our preliminary investigations extending up to 14 days post-infection revealed a progression consistent with our seven-day observations, with extensive cellular damage and widespread infection. This suggests that our in vitro model reflects infection dynamics in susceptible hosts rather than tolerant or resistant species. While extended observation periods could offer additional insights, the severe cellular pathology observed beyond seven days suggests that this model may be less suited for studying prolonged host-pathogen interactions.

This was addressed in the discussion at lines 420-429: Our in vitro model closely reflects the rapid and severe disease progression observed in hypersusceptible species, where Bsal proliferates unchecked, leading to host cell destruction within a timeframe consistent with in vivo observations in fire salamanders. Based on our own data extending up to 14 days post-infection, prolonging the observation period only reinforces this trajectory, with no indication of persistence or recovery. Studying prolonged infection dynamics, including persistence or potential host recovery, would possibly require alternative cell models. Future efforts to establish keratinocyte-derived caudatan cell lines, particularly those with differing β-galactose content, could provide valuable insights into species-specific differences in susceptibility and could allow for a more nuanced investigation of chronic infections.

Comment 5: The current model does not account for potential host-specific variations, such as differences in susceptibility among various amphibian species. Expanding the study to include multiple amphibian cell lines could enhance its applicability.

Response 5: Indeed, Bsal susceptibility varies among amphibian species and even among individuals. Hypersusceptible species, such as the fire salamander (Salamandra salamandra), invariably succumb to infection even when exposed to very low numbers of Bsal zoospores, failing to mount any significant protective response (Stegen et al., 2017). In contrast, other species, exhibit more variable and dose-dependent responses, ranging from self-limiting infections with spontaneous clearance to lethal disease.

Host susceptibility has been linked to the skin glycosylation patterns of salamanders, with cutaneous galactose content serving as a key predictor of Bsal colonization intensity across species. However, while colonization is an important step in infection, galactose content alone does not reliably predict disease progression. Extensive colonization does not necessarily lead to mortality, as the outcome is ultimately shaped by a combination of innate and adaptive immune responses and environmental factors.

We agree that incorporating multiple caudata epithelial cell lines, particularly those with varying β-galactose content, could provide additional insights into species-specific susceptibility. This would allow for a more detailed examination of early host-pathogen interactions. However, the subsequent disease outcome is influenced by numerous host and environmental factors that extend beyond the scope of this in vitro model.

This was addressed in the discussion at lines 412-419: Bsal susceptibility varies widely among amphibians, from hypersusceptible species such as fire salamanders, which fail to mount a protective response and rapidly succumb to infection, to species exhibiting variable responses [30]. Host susceptibility has been linked to skin glycosylation patterns, with cutaneous galactose content correlating with Bsal colonization intensity [18]. However, while colonization is an important step in infection, it does not necessarily predict disease outcome. Extensive colonization does not always lead to mortality, as the infection trajectory is ultimately shaped by a combination of innate and adaptive immune responses and environmental factors.

Comment 6: Figure 3 (lines 332-339), for a reader who is not familiar with the morphology will find this hard to understand. Maybe an illustration, highlighting the cell morphology is required. The abbreviation "GT" is missing in the caption.

Response 6: Illustration: To improve clarity for readers unfamiliar with the morphology, we have created an overview figure displaying HE, GMS, and fluorescent staining of Bsal-infected fire salamander skin tissues. This supplementary figure (Figure S4) highlights the morphology and key structures to aid understanding. Additionally, the abbreviation 'GT' has been added to the figure caption.

Figure 3. Temporal dynamics in early Bsal host interactions. Composite image depicting early interactions of Bsal with the stratum corneum of FTE explants from fire salamanders that were exposed to Bsal zoospores for (a) 2, (b) 4 or (c-e) 24 hours [22], using (a,d) GMS and (b,c,e) fluorescent stainings. (a) Two hours after inoculation, attachment to the stratum corneum is evident. (b) Subsequently, at the 4-hour time point, distinct cell wall thickening (*) indicates encystment of Bsal zoospores. By 24 hours p.i., there is evident active penetration of the stratum corneum, typified by (c-d) germ tube (GT)-mediated invasion and (e) intracellular content transfer. Scale bar =10 µM. For a detailed overview of the skin morphology and key structures, see Figure S4.

Figure S4. (a) HE staining, (b) GMS staining, and (c) fluorescent staining addressing the skin morphology of fire salamander skin tissue. The epidermis includes (1) the stratum corneum, which serves as the attachment site for Bsal spores (2). These spores penetrate deeper epidermal layers, including (3) the stratum granulosum, stratum spinosum, and stratum germinativum. The basal lamina (4) separates the epidermis from the underlying dermis, which consists of loose connective tissue (5) and contains melanin granules appearing as dark spots (6) in the dermis and extending towards the epidermis. Beneath the dermis, a well-defined muscle layer (7) provides structural support and mobility. SB = 10 µM

Reviewer 2 Report

The study presents a well-structured in vitro model for studying Batrachochytrium salamandrivorans (Bsal) infection using A6 cells. However, clarity and detail in methodological choices and data presentation can be improved to enhance reproducibility and scientific rigor.

1. The abstract presents a concise summary, but it could benefit from including key quantitative findings to provide a clearer overview of the study's impact.

2. Ethical considerations regarding the use of alternative models are well-addressed, but more discussion on the limitations of in vitro models compared to in vivo studies would enhance the introduction.

3. Lines 67-75: Specify the origin of the Bsal strain used. Were any additional measures taken to verify the purity of the culture beyond filtration and microscopy?

4. Lines 76-85: The rationale behind the progressive change in media composition (from 30% to 70% L-15) should be clarified. How does this transition influence fungal adherence and growth dynamics?

5. Line 97: What criteria were used to determine the exposure duration to cytochalasin D for assessing Bsal and A6 cell viability?

6. Line 141: The salamander skin control used is valid, but were these tissues processed identically to A6 cells to ensure consistency in staining results?

7. Lines 169-184: The manuscript describes actin cytoskeleton rearrangements; however, were any quantitative fluorescence intensity measurements performed to support visual observations?

8. Lines 245-246: The description of "widespread host cell death" could be strengthened with quantitative data such as cell viability percentages or counts.

9. Line 295: Was any attempt made to rescue actin polymerization after cytochalasin D treatment to confirm the finding?

10. Lines 303-308: The results suggest a close parallel with in vivo infection; however, were infection rates or severity scores compared quantitatively across models?

11. Lines 248-250: The observation of both endobiotic and epibiotic growth should include quantitative data on the proportion of each growth type observed over time.

Author Response

Response to Reviewer Comment

Comment 1: The abstract presents a concise summary, but it could benefit from including key quantitative findings to provide a clearer overview of the study's impact.

Response 1: We appreciate the reviewer’s input regarding the quantitative analyses. In response, we have incorporated additional quantitative findings into the text to provide a clearer overview of the study's impact. We have also addressed the extra quantitative analysis in the sections below and made the necessary adjustments to where appropriate.

Comment 2: Ethical considerations regarding the use of alternative models are well-addressed, but more discussion on the limitations of in vitro models compared to in vivo studies would enhance the introduction.

Response 2: In response, we have briefly elaborated on this aspect within the introduction to provide a clearer context. Additionally, we have expanded upon these limitations in the discussion section, where we specifically address the drawbacks of our in vitro model in relation to in vivo studies.

Introduction Lines 35-41: These studies offer a comprehensive understanding of disease progression within the complex physiological and immunological context of the host organism. In vivo models are particularly advantageous for studying species-specific and individual variations in responses to Bsal infection, as they account for factors such as amphibian immunity and environmental conditions that play a crucial role in disease development [6-7]. They provide a more holistic understanding and they also allow for the evaluation of long-term effects.

Discussion Lines 412-449: Bsal susceptibility varies widely among amphibians, from hypersusceptible species such as fire salamanders, which fail to mount a protective response and rapidly succumb to infection, to species exhibiting variable responses [30]. Host susceptibility has been linked to skin glycosylation patterns, with cutaneous galactose content correlating with Bsal colonization intensity [18]. However, while colonization is an important step in infection, it does not necessarily predict disease outcome. Extensive colonization does not always lead to mortality, as the infection trajectory is ultimately shaped by a combination of innate and adaptive immune responses and environmental factors. Our in vitro model closely reflects the rapid and severe disease progression observed in hypersusceptible species, where Bsal proliferates unchecked, leading to host cell destruction within a timeframe consistent with in vivo observations in fire salamanders. Based on our own data extending up to 14 days post-infection, prolonging the observation period only reinforces this trajectory, with no indication of persistence or recovery. Studying prolonged infection dynamics, including persistence or potential host recovery, would possibly require alternative cell models. Future efforts to establish keratinocyte-derived caudatan cell lines, particularly those with differing β-galactose content, could provide valuable insights into species-specific differences in susceptibility and could allow for a more nuanced investigation of chronic infections.

However, colonization alone does not necessarily predict disease outcome, as innate and acquired immune responses and environmental factors ultimately shape infection progression [18]. Amphibian immune mechanisms are critical to understanding Bsal infection. The amphibian skin plays a central role in mucosal immunity, with the microbiome and antimicrobial peptides providing protection against pathogens, including Bsal [6,12]. In terms of innate and adaptive immune responses, Bsal has been shown to suppress these defenses in infected salamanders, impairing their ability to mount an effective response, suggesting immune-dampening properties [31]. As our study focused primarily on Bsal-host cell interactions at the cellular level, it did not account for the role of host immunity or pathogen adaptation. Incorporating immune components, such as conditioned media containing antimicrobial peptides or even co-culture systems with immune cells, could provide a more comprehensive picture of host-pathogen interactions in future studies. Furthermore, while the controlled nature of in vitro systems allows for precise dissection of specific host-pathogen interactions, it does not replicate the fluctuating environmental conditions that influence Bsal infectivity and survival in natural settings. Temperature, humidity, salinity, and even substrate are environmental factors known to influence amphibian disease outcomes [7,32-34]. Future applications of this model could involve controlled environmental manipulations to assess how these factors influence Bsal-host interactions, bridging the gap between laboratory-based findings and ecological relevance.

Comment 3: Lines 67-75: Specify the origin of the Bsal strain used. Were any additional measures taken to verify the purity of the culture beyond filtration and microscopy?

Response 3: The Bsal type strain (AMFP 13/01) was the first Bsal chytrid isolated from a Bsal-positive fire salamander (Salamandra salamandra) in the Bunderbos, the Netherlands, in 2013. The isolation and characterization were described in reference [2], which is cited after the type strain in the Materials and Methods section.

To make this more clear to the readers, we added the details of origin as well at Lines 74-75: The Bsal type strain (AMFP 13/01), originating from a Bsal-positive fire salamander (Salamandra salamandra) [2]...

Our Bsal stock culture, is free from bacterial contamination. Once the culture is sporulating, by using a pore size 10 µM filter, we filter out the zoospores and remove the bigger sporangia. This was done exactly as mentioned in the materials and methods section. No additional measures were taken since this is an standardized and widely used technique to collect zoospores as explained in other references as well:

DOI: 10.1002/edn3.86

DOI: 10.1038/s41598-020-73800-y

DOI: 10.1038/s41467-020-19241-7

DOI: 10.1371/journal.pone.0225224

Comment 4: Lines 76-85: The rationale behind the progressive change in media composition (from 30% to 70% L-15) should be clarified. How does this transition influence fungal adherence and growth dynamics?

Response 4: Bsal is sensitive to in vitro conditions, particularly media composition. While it thrives in TGhL medium, its growth can be influenced in nutrient-rich cell culture media. Preliminary experiments assessed the impact of PBS and varying concentrations of L-15 medium on Bsal, focusing on zoospore motility and viability. Results indicated that zoospores retained motility in L-15 concentrations up to 30%, but motility decreased at concentrations above 40%. Given that motility is crucial for infection, the initial contact phase was conducted in 30% L-15 medium to preserve this trait. However, prolonged exposure to low-concentration L-15 medium led to the formation of empty sporangia, suggesting fungal stress. Additionally, A6 cells require higher L-15 concentrations for optimal growth. Based on prior research indicating that Bsal attachment occurs within the first two hours, the medium was subsequently adjusted to 70% L-15. This concentration supports both Bsal development and A6 cell health, providing a balanced environment for infection studies.

We tried to make this more clear to the readers by giving more info at Lines 92-95: This strategic modulation of media composition was implemented to preserve Bsal zoospore motility during the critical initial attachment phase and to promote subsequent fungal development, while simultaneously maintaining the health and viability of the A6 host cells throughout the infection process.

Comments 5: Line 97: What criteria were used to determine the exposure duration to cytochalasin D for assessing Bsal and A6 cell viability?

Response 5: We selected a 24-hour exposure to cytochalasin D (CD) to assess Bsal and A6 cell viability based on several considerations. Studies on Bd have shown that the fungus adheres to and internalizes into host cells within the first 24 hours (Verbrugghe et al., 2019). Given the similarities between Bd and Bsal, we anticipated comparable behavior for Bsal. Supporting this, ex vivo and in vivo studies on Bsal have demonstrated adherence and internalization within a similar timeframe (Li et al., 2021). CD disrupts the actin cytoskeleton, which is crucial for processes like endocytosis. In studies involving other eukaryotic cells, CD has been used effectively to investigate the role of the actin cytoskeleton in cellular processes (Dalle et al., 2010; Wächtler et al., 2012). By choosing a 24-hour exposure to CD, we aimed to effectively disrupt the actin cytoskeleton to assess its role in the initial Bsal infection step.

We tried to make this more clear to the readers by giving more info at Lines 106-107: A 24-hour exposure to CD was chosen to encompass the initial invasion step of Bsal infection, allowing us to assess the impact of actin disruption during the early stages of infection

Comments 6: Line 141: The salamander skin control used is valid, but were these tissues processed identically to A6 cells to ensure consistency in staining results?

Response 6: To address the reviewer's concern regarding the consistency of processing between the salamander skin control tissues and the A6 cells, we confirm that the staining protocol was identical for both, involving a 30-minute incubation with Ricinus communis agglutinin I (RCA I). The primary difference lies in the preparation of the tissue samples, which were paraffin-embedded. Consequently, these samples underwent deparaffinization, antigen retrieval, and blocking steps prior to staining, as detailed in Wang et al. (2021). In the manuscript of Wang et al. Bouin fixation is described, but formol fixation (4%) gives the same results.

Comments 7: Lines 169-184: The manuscript describes actin cytoskeleton rearrangements; however, were any quantitative fluorescence intensity measurements performed to support visual observations?

Response 7: No, quantitative fluorescence intensity measurements were not performed in this study to support the visual observations of actin cytoskeleton rearrangements. Our analysis was primarily based on qualitative observations of actin cocoon formation, which were visually assessed through microscopy. The goal of this manuscript is to present the Bsal in vitro infection model as a valuable tool for studying host-pathogen interactions, particularly as it offers a way to explore these dynamics without the need for extensive animal experimentation. We hope that this will inspire further research in the Bsal field, and future studies could incorporate quantitative measurements of fluorescence intensity to deepen the understanding of cytoskeletal rearrangements and other cellular processes.

Comments 8: Lines 245-246: The description of "widespread host cell death" could be strengthened with quantitative data such as cell viability percentages or counts.

Response 8: We had previously assessed cell viability seven days post-Bsal infection using a neutral red assay; however, these data were not originally included in the manuscript. In response to the reviewer’s suggestion, we have now added the results to the supplementary material to provide quantitative support for the description of 'widespread host cell death'.

Materials and methods L 168: Cell viability 7 days p. i. was assessed using neutral red [21]. 

Figure S1. Effect of Bsal infection on the viability of A6 cells, 7 days post infection. Relative viability (%) of A6 cells infected with Bsal, 7 days p. i., determined by neutral red assay. NC = uninfected A6 cells (negative control); Triton_C = A6 cells treated with 1% triton X-100 (positive control); Bsal = Bsal-infected A6 cells. *** indicates a significant difference toward the NC group with p < 0.001. Statistical significance was assessed using a generalized linear mixed model (GLMM) with a beta distribution, where condition was the fixed effect and random intercepts were specified for the nested structure of technical replicate/biological replicate.

Comments 9: Line 295: Was any attempt made to rescue actin polymerization after cytochalasin D treatment to confirm the finding?

Response 9: We observed continued Bsal invasion even after cytochalasin D treatment, indicating that actin polymerization inhibition did not entirely prevent fungal invasion. Therefore, rescuing actin polymerization would not have influenced the observed outcome, as invasion persisted even after a 24h treatment. To investigate the potential for actin recovery, we performed a washout procedure following the 24-hour cytochalasin D treatment, which, in principle, represents an attempt to rescue actin polymerization.

Comments 10: Lines 303-308: The results suggest a close parallel with in vivo infection; however, were infection rates or severity scores compared quantitatively across models?

Response 10: The infection rates or severity scores were not quantitatively compared across models due to the inherent differences between the systems used. Amphibian species exhibit considerable variability in their response to Bsal, with factors such as the presence of B-galactose on the skin playing a key role in initial attachment. However, the subsequent development of infection and severity (including mortality) is influenced by numerous additional factors, such as immune response, environmental conditions, and infection dose.

All tissue slides used in this study originated from previous research with fire salamanders, but the aforementioned factors cannot always be strictly controlled or standardized across in vivo, ex vivo, and in vitro models, making direct quantitative comparison challenging. Therefore, while our study provides important insights into early host-pathogen interactions, we recognize that comparisons across systems involve trade-offs, especially when attempting to extrapolate findings from ex vivo and in vitro models to more complex in vivo situations.

Regarding the issue of quantification, in vivo and ex vivo systems can indeed be quantitatively assessed using qPCR, which remains the gold standard for infection quantification, as demonstrated in previous studies. However, quantification in the in vitro system is more complicated due to the interference of epibiotic growth with qPCR data. This epibiotic growth complicates the interpretation of fungal load, potentially leading to misleading estimates of the actual infection burden. This challenge remains an ongoing issue for in vitro models, further complicating direct comparisons with the in vivo and ex vivo data.

We tried to make this more clear to the readers by giving more info at Lines 450-463: In this study, we focused on early host-pathogen interactions using tissue slides from fire salamanders, based on previous in vivo and ex vivo work [18,23-24]. However, due to inherent biological differences across models, direct quantitative comparisons of infection rates or severity scores between in vivo, ex vivo, and in vitro systems are challenging. Factors such as species variability in response to Bsal, immune differences, environmental conditions, and infection dose all contribute to the complexity of infection progression and severity [6-7,30]. While in vivo and ex vivo models can be quantitatively assessed using qPCR, this approach is complicated in our in vitro system due to the interference of epibiotic growth, which confounds accurate fungal load measurement. Numerous pathogens, such as Salmonella, benefit from established in vitro assays like the gentamicin protection assay, which enables accurate quantification of intracellular bacteria [35]. Optimizing a similar tool with our in vitro model could facilitate direct intracellular quantification of Bsal using qPCR or EMA-qPCR, improving our ability to assess fungal load more precisely [36].

Comments 12: Lines 248-250: The observation of both endobiotic and epibiotic growth should include quantitative data on the proportion of each growth type observed over time.

Response 12: To accurately quantify the proportion of endobiotic and epibiotic growth types observed over time, we recommend using an (EMA) qPCR technique applied to the entire well. This would provide a more precise measure of the two growth types, but then we would need an antifungal agent that does not penetrate host cells and which can kill Bsal. So in principle, a method similar to the gentamicin protection assay used for bacteria. However, at present, such a technique is not available yet for Bsal systems.

In the absence of this, imaging-based techniques such as microscopy can be employed to assess the distribution of Bsal; however, these methods are limited as they typically do not cover the entire well. Additionally, time-lapse imaging presents challenges due to the initial extracellular presence of Bsal, which can transition into intracellular growth over time. At early time points, the presence of extracellular material complicates quantification, as it may represent intracellular growth, epibiotic growth, or a remaining fraction of the mother thallus, making it difficult to distinguish between these different growth forms with certainty.

Thus, epibiotic growth can generally be detected with certainty from day 3 p.i. onwards, as this is when extracellular development of Bsal becomes more distinct. However, from day 7 p.i. onward, host cell lysis and the subsequent release of fungal material can occur, making it difficult to distinguish between epibiotic growth and the release of intracellular material. Once spores are released, they can re-adhere and encyst, eventually staining with CFW, further complicating the interpretation of growth types. In our opinion, the most appropriate time points to estimate the difference between endobiotic and epibiotic growth would be after the complete internalization of Bsal and before the dispersal of newly formed spores.

Therefore, we used imageJ to determine the proportion at timepoint day 3 post infection to give the readers more information.

Materials L182-186: To determine the proportion of intracellular (endobiotic) and extracellular (epibiotic) growth of Bsal 3 days p.i., three biological replicates were performed, each with at least three technical replicates and two imaged regions per replicate. ImageJ software was used for image analysis, applying a threshold to each channel followed by particle analysis to quantify the total area.

L270-271: At day 3 p. i., the proportion of extracellular Bsal was 49.99% ± 8.91 (SEM) across all biological and technical replicates.

Round 2

Reviewer 1 Report

The authors have addressed the comments/concerns I had in the previous version. I accept the current version that has been submitted. 

Thank you.

NA